# How value perspectives influence decision-making in the South African private healthcare sector: A cross-sectional comparative study

Anchen Laubscher[1,2]*, Reitze N. Rodseth[1,3], Francois Retief[4], Adrian Saville[2]

**1** Netcare Ltd, Johannesburg, South Africa, **2** Gordon Institute of Business Science (GIBS), University of Pretoria, Sandton, South Africa, **3** Department of Anaesthesiology and Critical Care, Nelson R. Mandela School of Medicine, University of KwaZulu-Natal, Durban, South Africa, **4** Department of Obstetrics and Gynaecology, Faculty of Medicine, University of Pretoria, Pretoria, South Africa

* anchen.laubscher@netcare.co.za

## Abstract

**Data Availability Statement:** The data underlying the results presented in this study are available from the Figshare repository (DOI: https://doi.org/10.6084/m9.figshare.27693426.v1).

### Background

Every healthcare clinical event aims to create value at a certain cost. This value has been defined as the outcome achieved (the degree to which a care event achieved a clinical goal) divided by the cost incurred (determined by the combined price charged by the care provides) to generate the outcome. Subsequently, patient experience has been included as a third factor contributing to value of care, but its value and relationship relative to clinical outcome and event cost is not well understood. This cross-sectional comparative study explored the relative importance of 1) clinical outcome, 2) event cost, and 3) patient experience as they relate to the value of care in the South African private healthcare context.

### Materials and methods

Using a value perspectives survey, healthcare consumers (n = 662) and healthcare providers (n = 318) distributed 100 points between the three factors according to how they perceived their value. They were then asked to assess the value of the three factors across six clinical scenarios progressing in clinical severity.

### Results

For all scenarios, all participants valued patient experience above event cost, but lower than clinical outcome. However, there were significant differences between consumers and providers in the relative value assigned to each of the three factors. These values changed as the severity of the surgical and medical scenarios changed. Patient experience was consistently assigned a higher value than event cost, thereby making a strong argument for its inclusion into the healthcare value equation.

**Funding:** The author(s) received no specific funding for this work.

**Competing interests:** AL and RR are employees of Netcare Limited, a private hospital group in South Africa. FR and AS declare that they have no competing interests.

**Abbreviations:** ANOVA, Analyses of variance; CI, Confidence interval; SD, Standard deviation; VCI, Value Care Index; ZAR, South Africa Rand.

## Conclusion

Both South African healthcare consumers and providers assigned significant value to patient experience across a range of clinical scenarios. These findings suggest that patient experience should be included as a factor in the Value Care Index (VCI) where VCI = (Outcome ÷ Cost) x Patient Experience.

## Introduction

Every healthcare clinical event aims to create value at a certain cost. More specifically, value in health care is the outcome achieved (the degree to which a care event achieved a clinical goal) divided by the cost incurred (determined by the combined price charged by the care provides) to generate the outcome. This can be expressed in the value equation where Value of Care = (Outcome ÷ Cost) [1].

While outcome and cost undoubtedly contribute to value of care it is argued that value is heavily influenced by other factors, key among which is patient experience [2, 3]. Patient's experience has been understudied and underutilised in value creation and value based care strategies in healthcare [2].

Berwick et al introduced the concept of the 'Triple Aim', which includes the elements of care, health, and cost, where care refers to the patient's subjective experience of the care episode [4]. Damberg et al also described consumer directed healthcare as being important in value creation strategies; the emphasis again falling on the patient's expectations and experience of care delivered experience [2]. Damberg explained how patient's experience in the healthcare context is understudied, whereby a value-based payment study conducted in 2014 found that only 17% of value-based care programmes measured goals that related to the patient's experience or the patient's value perspective of the care delivered.

Porter highlights the need for healthcare systems to focus on clinical outcomes and cost of clinical event but acknowledges that healthcare systems often fail to recognise a fundamental criterion for health services excellence and value creation, namely whether patient expectations of care are met [1]. If patient's experience is not considered a factor in healthcare value creation strategies, clinical events in healthcare delivery may fail to create value for the patient even though it appeared that cost and outcome goals were achieved.

While other healthcare valuation models have focused on including patient safety or the risk of adverse event occurrence, the centrality of the patient's subjective experience of their care episode was the impetus for us to study this valuation model.

The study, therefore, aimed to determine the need to add patient's experience to the healthcare value equation as originally described, to understand the relative importance of this additional factor in relation to other factors, and to examine how this relative importance might differ between consumer and provider groups.

Further to exploring the role of patient's experience in value of care, the purpose of the study is to obtain a deeper understanding of the differences in value perspectives between healthcare consumer and provider groups. Value in the healthcare context remains an abstract construct influenced by perception that can vary between different stakeholders depending on the role each of them plays in the care event [1]. Multiple stakeholders are at play in healthcare delivery. Key stakeholders include the consumer or patient and provider or clinician. Often-times, stakeholders in healthcare delivery have conflicting goals, leading to divergent approaches and taking away from the shared value agenda event [1, 5]. A deeper

understanding of the differences in value perspectives that inform decision-making in the healthcare context is therefore required.

This study, a quantitative review of primary data, aims to gain insight into the value perspectives of consumers and providers in the South African private healthcare context and to examine if these value perspectives differ between the two groups. Both providers and consumers from the South African private sector were included in this study. Consumers refer to all privately paying or medically-insured patients in the South African healthcare context, thus the consumers making use of private healthcare. Providers refer to clinicians (medical doctors) registered to work in private healthcare practice in South Africa, thus actively practicing or having actively practiced clinical medicine in the private healthcare sector.

The study explored the differences between the relative importance of the three factors (clinical outcome, cost of clinical event, and patient's experience) that make up the value of care for both consumers and providers, in general and specifically in terms of medical and surgical procedures. Patient's experience is compared with clinical outcome and cost of clinical event in a series of scenarios aimed at determining the relative importance or weighting attributed to each of the three factors in the healthcare value equation to inform a better understanding of the differences in value perspectives prevalent that, in turn, inform decision-making.

The objective of this study was to 1) explore patient experience's importance relative to clinical outcomes and event cost, across a range of clinical scenarios in the South African private healthcare context; 2) understand how relative importance between the three factors might differ between consumer and provider groups as the clinical severity of the scenarios changed and 3) explored the feasibility expressing the relationship between patient experience and the value equation as the Value Care Index (VCI): VCI = (outcome cost ÷ cost) x patient experience.

## Materials and methods

This study was a non-experimental, cross-sectional, comparative study between healthcare consumer and provider groups within the private healthcare sector in South Africa. The industry in which the study was conducted was the South African healthcare sector, specifically, the private sector where payment is made for medical services rendered. Clinicians are not employed by private healthcare institutions and operate as independent practitioners in these facilities, as dictated by the Health Professionals Council of South Africa (HPCSA). Patients are either insured by subscribing to and purchasing of medical aid cover or they pay for private medical care by means of cash at point of care. Medical aid cover ranges in benefit structures from all-inclusive to very limited packages, resulting in varied additional out-of-pocket payment implications when care is received. A non-probability sampling technique was used to collect data and included convenience, purposive and snowball sampling strategies [6]. The first author (AL) utilised her professional and personal network to collect data from the two sample groups using various social media platforms. The purposive strategy was employed to ensure a high number of clinician participants across a wide spectrum of subspecialties and registration categories with the HPCSA. Included in the sample were doctors working in both administrative and clinical capacities. Snowball sampling was used to leverage the networks available to the researcher. Participants were asked to distribute the survey to other potentially willing individuals both when they received an invitation and following completion of the survey.

Data were collected through a self-administered online survey, the Value Perspectives Survey, using the survey platform Qualtrics (full survey available in Appendix 1 in S1 File) and were collected from 1 to 30 September 2017. The survey link directed the participant to a cover

**Table 1. Details of the clinical scenarios presented to study participants.**

| Scenario | Description |
|---|---|
| Surgical minor severity | You require admission to a day ward for surgery to remove a skin tag. |
| Surgical moderate severity | You require admission to a general ward for surgery to repair a hernia in your groin. |
| Surgical major severity | You require admission to an intensive care unit for abdominal surgery for the treatment of liver cancer that has spread to other organs in your body. |
| Medical minor severity | You require admission to a day ward for control of your blood sugar levels that are too high. |
| Medical moderate severity | You require admission to a general ward to receive oxygen therapy for pneumonia. |
| Medical major severity | You require admission to an intensive care unit for a heart attack with severe shock as complication. |

letter after which consent was obtained. Participants were then directed to either the patient or doctor section of the survey. To ensure a between-subjects design, a participant could not compete the survey from both consumer and provider perspectives. Should a participant work in the healthcare industry as a healthcare worker in any category other than that of a doctor, they were required to complete the survey from a patient perspective.

After obtaining basic demographic information the survey presented clinical scenarios aimed at examining the participants' value perspectives—the primary focus of this study. They were asked to distribute 100 points between clinical outcomes, event cost, and patient experience according to their perceived relative importance. For example, a participant evaluating a clinical scenario where they were undergoing surgery for a hernia repair could decide to allocate 30 of the available 100 points to clinical outcome, 50 points to event cost, and 20 to patient experience. Clinical outcome was defined as the degree to which the clinical event achieved a clinical goal, event cost as the total price charged by the care providers, and patient experience as the degree to which the patient's expectation was met. In the first question participant were asked to provide a general perspective on the relative value of the three factors. After this general baseline question three medical and three surgical scenarios, which progressed in clinical severity, were described. For every question, the three factors were presented in random order. Scenarios were selected to include conditions that could be experienced by patients and encountered by clinicians across different age groups and genders. The scenarios used are described in Table 1.

The scenario questions were designed to indicate the progressive severity of the disease process. For surgical cases—from minor skin flap, to groin hernia, to cancer; and for medical cases—from high blood sugar, to pneumonia, to heart attack with septic shock. Further, the location where case was to be provided was also graded to indicate progression in disease severity—from day ward, to general ward, to intensive care unit.

Upon ethics approval, the survey was piloted on 15 participants using Google forms, and following feedback, was amended and transferred onto Qualtrics where it was piloted on another sample of 30 participants. Minor design-related adjustments were made, and the survey was distributed for data collection for the main study.

## Data analysis

Descriptive statistics were used to characterise the study sample. Tests of mean differences i.e., independent samples t-test, was used to assess whether value perspective differences existed

between consumer and provider groups and analyses of variance (ANOVA) tests were used to assess value perspective differences such as the severity of surgical and medical scenarios increased. Parametric assumptions were evaluated using Kolmogarov-Smirnov and Shapiro-Wilk tests. To examine the association between these variables of interest, univariate analyses using Pearson product-moment correlations were conducted. Independent samples t-tests were conducted to assess whether provider and consumer groups weighted value perspectives differently across the general value baseline and the surgical, and medical scenarios.

Cohen's d effect sizes were calculated to determine practical differences between the weighted perspectives between provider and consumer groups, with effect size classified as follows: small (d = 0.2), medium (d = 0.5) and large ($\geq$ 0.8). One way between-groups ANOVA tests were conducted to determine whether differences existed in value perspectives across severity of surgical and medical scenarios for both consumer and provider groups. The effect size for these analyses using ANOVA were calculated using eta squared ($\eta^2$), with effect size classified as follows: small ($\eta^2 = 0.01$), medium ($\eta^2 = 0.06$) and large ($\eta^2 = 0.14$. Data were analysed using IBM SPSS version 24 and alpha was set at 5%.

## Ethical considerations

Data collection, specifically in the healthcare sector, requires ethical practises, including respect for the participants, as well as transparency of what is being researched and confidentiality of data. Ethical clearance was obtained from the Gordon Institute of Business Science Research Ethics Committee and the University of Pretoria Human Research Ethics Committee (Temp2017-00651, 18/07/2017) before any research was conducted.

Participants were made aware that the results may be presented publicly and published in an academic journal. Furthermore, the results of the survey were kept private, and confidentiality was upheld throughout. Data were analysed and reported on in an anonymised, aggregated manner and the researcher ensured that participants could by no means be identified as no identifying information (name, ID etc.) was requested in the survey.

Individuals wanting to participate in this study needed to do so voluntarily. After receiving information on the nature and objective of the study, participation required indicating consent on the electronic questionnaire, which stipulated the voluntary nature of the study. There were no benefits or harm to participating in this study and participants could stop at any time without negative consequences.

When conducting research involving humans it is important to note ethical concerns that may arise during the execution of the research study. Throughout the research study the principal researcher ensured that ethical standards were upheld.

## Results

### Demographics

The study included a total of 1043 participants between 19 to 88 years with a mean age of 44 years (standard deviation [SD] 11.8) of which 58% (n = 605) were female. Private healthcare consumers comprised 662 participants and providers comprised 381 participants. A total of 298 individuals were excluded from the study for various reasons including non-consent, incomplete survey responses, consumers who utilised, and providers who practiced in public healthcare only, and providers only practicing outside of South Africa. Where data were missing for specific fields, these patients were excluded from that analysis. For the most part, the data was found to be normally distributed and parametric techniques were employed.

In the consumer group 70.8% (n = 469) were female; 91.5% (n = 606) had an education level beyond grade 12, and 47.7% (n = 316) had a gross monthly income < South African

Rand (ZAR) 50 000. The majority (88.4%, n = 585) only made use of private healthcare, while the remainder used a combination of private and public; 94.4% (n = 625) were members of a medical aid, of which 85.8% (n = 536) had been so for more than 10 years. Consumers contributed an average of ZAR 4722 (SD = 2792) per month for medical aid services.

In the provider group 35.7% (n = 136) were female and most practiced in South Africa (94.2%, n = 359). Specialists comprised 64.6% (n = 246) of the group and the majority worked in clinical medicine (89%, n = 339) with the remainder in medical administration.

Of those working in clinical medicine, 25.4% focused on medical, 49% on surgical, and 3.5% on diagnostic practices.

## Overall value perspectives between consumers and providers

Across all categories (general baseline, surgical and medical scenarios), clinical outcome was allocated the highest value for both consumer and provider groups, followed by patient experience and then event cost (Fig 1). For clinical outcome, all three categories were positively correlated with each other, while it was negatively correlated with both cost and patient experience. Results were found to be similar across consumer and provider groups, with only marginal differences. The Pearson correlation matrix can be found in Appendix 2 in S1 File.

## Relative weighting of value perspectives between consumers and providers

When comparing the relative weight between the two groups, in the general baseline category, consumers valued event cost higher than providers (22.8 vs. 17.6; p < 0.001, d = 0.39), while providers valued patient experience higher than consumers (28.1 vs. 25.0; p = 0.001, d = 0.21).

For the surgical scenario categories, consumers valued clinical outcome higher than providers (52.7 vs 45.4; p = 0.001, d = 0.43), while providers valued event cost (24.2 vs 20.9;

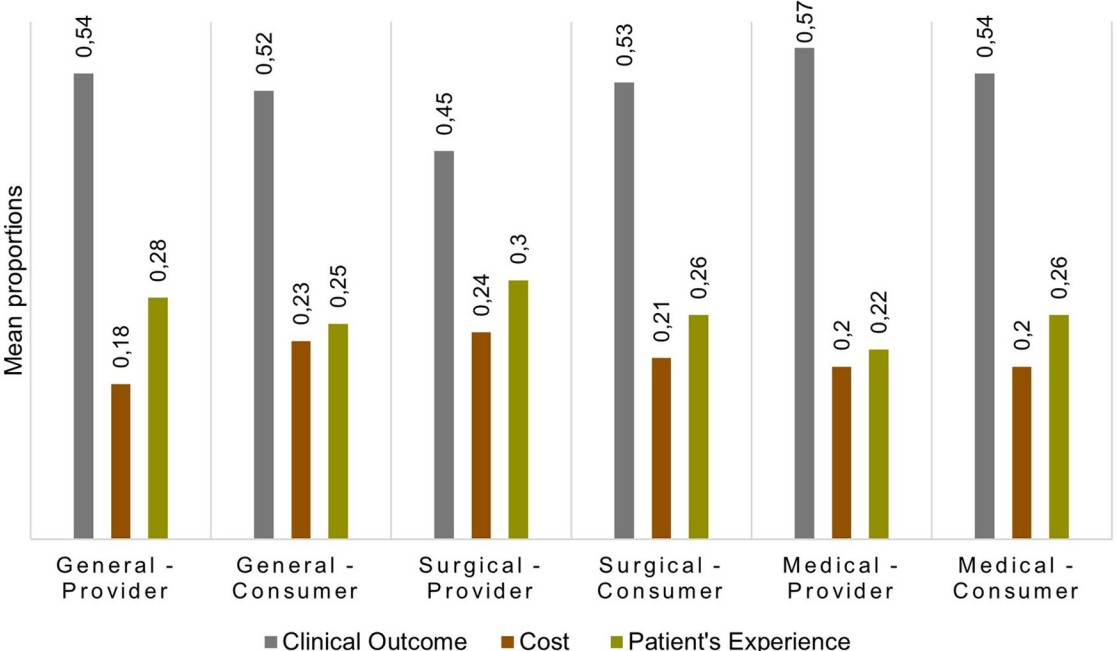

**Fig 1. Consumer and provider mean value perspectives expressed as proportions across general baseline, surgical, and medical scenario categories.**

**Table 2. The difference in value assigned to clinical outcome, event cost and patient experience between consumers and providers across three different clinical categories (general baseline, surgical and medical scenarios).**

| Category | Value factor | Consumer value weighting | Provider value weighting | P value | Cohen's d |
|---|---|---|---|---|---|
| | | Mean (SD) | Mean (SD) | | |
| General baseline | Clinical outcome | 52.2 (21.37) | 54.4 (16.71) | 0.073 | 0.11 |
| | Event cost | 22.8 (15.46) | 17.6 (10.84) | < 0.001* | 0.39 |
| | Patient experience | 25.0 (15.82) | 28.1 (13.40) | 0.001* | 0.21 |
| Surgical scenarios | Clinical outcome | 52.7 (18.62) | 45.4 (15.23) | < 0.001* | 0.43 |
| | Event cost | 20.9 (13.65) | 24.2 (12.67) | < 0.001* | 0.25 |
| | Patient experience | 26.4 (13.54) | 30.4 (10.87) | < 0.001* | 0.33 |
| Medical scenarios | Clinical outcome | 54.4 (18.77) | 57.2 (17.68) | 0.029* | 0.15 |
| | Event cost | 20.0 (13.46) | 20.1 (12.84) | 0.900 | 0.01 |
| | Patient experience | 25.6 (12.95) | 22.7 (10.76) | < 0.001* | 0.24 |

* Statistically significant using independent samples t- test,

SD = standard deviation.

p < 0.001, d = 0.25) and patient experience (30.4 vs 26.4; p < 0.001, d = 0.33) higher than consumers.

In the medical scenario categories, consumers valued patient experience higher than providers (25.6 vs 22.7; p < 0.001, d = 0.24), while providers valued clinical outcomes higher than consumers (57.2 vs 54.4; p < 0.001, d = 0.15). These results are detailed in Table 2.

## Impact of clinical severity on value perspectives between consumers and providers

As severity of the medical and surgical scenarios increased, consumers gave greater value to clinical outcome and less to cost and patient experience (Table 3). While statistically significant, the actual differences in mean scores between the groups were small to moderate. The effect size ($\eta^2$) ranged from 0.01 (small) for surgical patient experience to 0.09 (medium) for medical clinical outcomes. The same pattern was observed for providers (Table 4). Interestingly, when evaluating the surgical scenario category, providers valued clinical outcome highest in the moderate severity scenario and not in the most severe scenario. By contrast, when evaluating the medical scenario category, providers valued clinical outcome highest in the severe severity medical. This same pattern was also seen for patient experience, where providers assigned the greatest value to the moderate severity surgical scenario and not to the most severe scenario. This pattern was again different from that seen in the medical scenarios. Actual difference in mean scores between the groups were small to moderate despite reaching statistical significance, the effect size ($\eta^2$) ranging from 0.02 to 0.11.

## The value of patient experience

Patient experience was consistently valued higher than event cost, across all scenarios, thereby making a strong argument for its inclusion into the healthcare value equation. We propose that this relationship be expressed as the Value Care Index (VCI): VCI = Value of Care = (outcome ÷ cost) x patient experience, and that the mean value proportions can be used as coefficients and weightings in the equation.

**Table 3. Comparisons of surgical and medical severity value perspectives for consumers of private healthcare.**

| Scenario classification | Value factor | Mild severity | Moderate severity | Severe severity | p value | $\eta^2$ |
|---|---|---|---|---|---|---|
| | | Mean (CI) | Mean (CI)) | Mean (CI) | | |
| **Surgical** | Clinical Outcome | 46.0 (44.23–47.78) | 52.1 (50.33–53.87) | 60.0 (57.75–62.23) | < 0.001* | 0.06 |
| | Event cost | 25.34 (23.90–26.88) | 21.1 (19.80–22.49) | 16.3 (14.82–17.71) | < 0.001* | 0.04 |
| | Patient experience | 28.6 (27.20–30.01) | 26.8 (25.42–28.09) | 23.7 (22.06–25.43) | < 0.001* | 0.01 |
| **Medical** | Clinical Outcome | 47.3 (45.51–48.99) | 51.6 (49.90–53.38) | 64.34 (62.14–66.54) | < 0.001* | 0.09 |
| | Event cost | 23.6 (22.11–25.12) | 20.3 (19.00–21.61) | 16.0 (14.49–17.53) | < 0.001* | 0.03 |
| | Patient experience | 29.1 (27.71–30.56) | 28.1 (26.66–29.46) | 19.7 (18.16–21.15) | < 0.001* | 0.06 |

\* Statistically significant using analysis of variance,

CI = 95% confidence interval, $\eta^2$ = eta squared.

## Discussion

Value in the healthcare context remains an abstract construct influenced by perceptions that can vary between different stakeholders depending on the role each of them plays in the care event [3]. This study aimed to understand how healthcare consumers and providers relatively valued clinical outcome, patient experience, and event cost across a range of clinical scenarios in the South African private healthcare context.

### Relative weighting of value perspectives between consumers and providers

Consumers and providers, across all scenarios, valued clinical outcome the highest, followed by patient experience, and lastly event cost. This study supports many of the key assumptions underpinning the value equation. First, in support of Porter's work, it underscores clinical

**Table 4. Comparisons of surgical and medical severity value perspectives for providers of private healthcare.**

| Scenario classification | Value factor | Mild Severity | Moderate Severity | Severe severity | p value | $\eta^2$ |
|---|---|---|---|---|---|---|
| | | Mean (CI) | Mean (CI)) | Mean (CI) | | |
| Surgical | Clinical outcome | 39.0 (36.91–41.15) | 51.8 (49.92–53.8) | 45.4 (42.68–48.19) | < 0.001* | 0.06 |
| | Event cost | 28.2 (26.13–30.35) | 20.4 (18.87–21.8 | 23.9 (21.89–25.84) | < 0.001* | 0.03 |
| | Patient experience | 32.7 (30.95–34.5) | 27.8 (26.31–29.3) | 30.7 (28.71–32.69) | < 0.001* | 0.02 |
| Medical | Clinical outcome | 50.6 (48.35–52.75) | 54.1 (52.00–56.3) | 66.8 (64.43–69.24) | < 0.001* | 0.10 |
| | Event cost | 22.6 (20.90–24.31) | 20.4 (18.74–22.0) | 17.3 (15.50–19.08) | < 0.001* | 0.02 |
| | Patient experience | 26.8 (25.19–28.49) | 25.5 (24.00–27.0) | 15.9 (14.47–17.27) | < 0.001* | 0.11 |

\* Statistically significant using analysis of variance,

CI = 95% confidence interval, $\eta^2$ = eta squared.

outcome as being the key determinant in the value equation [1]. Second, with event cost receiving a consistently significant proportion of value, it affirms its importance in the value equation. However, event costs' low value relative to clinical outcome and patient experience highlights that value creation strategies in healthcare require more than mere cost containment [7].

## The value of patient experience

The study quantifies the importance of patient experience in the creation of healthcare value. Patient experience was consistently valued higher than event cost, across all scenarios, thereby making a strong argument for its inclusion into the healthcare value equation. We propose that this relationship be expressed as the Value Care Index (VCI): VCI = (outcome ÷ cost) x patient experience, and that the mean value proportions can be used as coefficients and weightings in the equation.

## Impact of clinical severity on value perspectives between consumers and providers

There are multiple stakeholders in healthcare delivery, and these often have conflicting goals [5, 8]. In this study consumers and providers both valued clinical outcome > patient experience > event cost. And generally, as the clinical scenarios became more severe, both groups assigned greater value to clinical outcome. However, relative to each other, there were significant differences in the value assigned to each factor. Consumers valued clinical outcome (surgical scenario), patient experience (medical scenario), and event cost (general baseline category) higher than providers. Providers valued clinical outcome (medical scenario), patient experience (general baseline category) and event cost (surgical scenario) higher than consumers. These results demonstrate that differences in value perspectives exist between consumers and providers and that this is affected by clinical severity. These findings provide healthcare managers, policymakers, and researchers within insight into these value differences and their influence on decision-making [8, 9]. The phenomenon presents itself as an interesting field for further research.

An interesting phenomenon was demonstrated in the array of surgical value perspectives in terms of event cost and clinical outcome where the highest value across scenarios was given to surgical cost in the mild severity example but the lowest given in the moderate severity scenario. This was different from the medical array of scenarios where the greatest value was given to the most severe scenario. Similarly, the highest value across scenarios for surgical clinical outcome was in the moderate severity category with the second highest in the severe category. This was again different from the result obtained in the medical array of scenarios. Of further importance was the finding that patient experience consistently held a greater perceived value as compared to event cost. This highlights the importance of centring the patient, and their experience of their care event, during all healthcare episodes.

The goal of any healthcare system should be to ensure that every person in the system has access to affordable, quality healthcare [10]. In South Africa, the inefficiency and poor quality of healthcare provided by the public sector has allowed the emergence of a competitive and expensive private sector [11, 12]. South Africa spends 41.8% of its total health expenditure on private, voluntary health insurance [13]. Still, only 17% of the South African population can afford private insurance and benefit from this disproportionate contribution.

Value creation and enhancement strategies in health are aimed at addressing efficiency and improving quality of care [14]. When embraced as a collective strategy these can serve to better align multiple stakeholders with divergent and even competing goals [8, 9]. Private healthcare

services, specifically, are expensive and reforms are required to ensure value creation strategies are implemented successfully, not only to contain cost but also to ensure clinical goals and patient's needs are met. Armed with an understanding of the value perspectives of consumers and providers specific to the South Africa economic context, the incentive should be to innovate towards delivering greater patient value [15]. These findings provide an opportunity to understand the requirements for the effective achievement of value-based care objectives, including cost control, quality improvement and better outcomes of care.

## Strengths

The observational nature of this study gives its findings broad applicability within the South Africa healthcare context. Its large sample size relative to the South African healthcare ensures that its findings will have significant relevance and external validity.

## Limitations

This study is limited in that the consumer sample may have included other healthcare practitioners such as nurses, paramedics, pharmacists, physiotherapists, occupational therapists, and dieticians, for example. However, as these healthcare practitioners will all at one stage be healthcare consumers, this is likely to increase the generalisability of the study findings. An additional limitation is that respondents, particularly non-healthcare practitioners, may not have personally experienced the clinical scenarios used in the questionnaire. However, the severity of the different categories was not specifically linked to the clinical scenarios described, but more broadly to the severity of those scenarios. The difference between being treated in a day ward, so compared to a general ward or intensive care would be apparent to non-healthcare professionals. Further, we were not able to explore the factors influencing value assignment in each group, thereby limiting the explanatory power of the study. Our findings are limited in that they relate to the specific population we studied. Therefore, care should be taken when interpreting the coefficients and weightings reported in this study. Similar studies conducted in different populations would assist in addressing this limitation.

## Conclusion

Healthcare should be united around the single overarching goal of providing patients with the highest possible value of care. This shared value agenda—specific to healthcare—is highly relevant in the South African context, but requires understanding from all stakeholders before its impact becomes apparent [16]. This value has been defined as the outcome achieved multiplied by the patient experience and then divided by the cost incurred. This cross-sectional comparative study explored the relative importance of 1) clinical outcome, 2) event cost, and 3) patient experience as they relate to the value of care in the South African private healthcare context. In this value perspectives survey, healthcare consumers and healthcare providers distributed 100 points between the three factors according to how they perceived their value. They were then asked to assess the value of the three factors across six clinical scenarios progressing in clinical severity.

For all scenarios, all participants valued patient experience above event cost, but lower than clinical outcome. However, there were significant differences between consumers and providers in the relative value assigned to each of the three factors. These values changed as the severity of the surgical and medical scenarios changed. Patient experience was consistently assigned a higher value than event cost, thereby making a strong argument for its inclusion into the healthcare value equation. These findings suggest that patient experience should be included

as a factor in the Value Care Index (VCI) where VCI = (Outcome ÷ Cost) x Patient Experience.

While our findings support the centrality of clinical outcomes as the key determinant in the value equation, they highlight the importance of patient experience over event cost and make the argument for its inclusion into the tradition value equation. Our study further contributes to our knowledge of this subject by describing how: 1) relative value proportions of clinical outcome, patient experience and patient event cost when determining healthcare value; 2) clinical severity impacts value assignment; 3) value judgements differ between medical and surgical clinical scenarios, and 4) healthcare consumers and providers differ in assigning value—all within the South African healthcare context.

It would be of value to repeat a similar study within different health care and socioeconomic contexts. It is likely that the perspectives of both health care providers and consumers would change based on the socioeconomic development level of their country as well as the availability and stability of that country's health care system. As such, our findings are specific to the South African private healthcare context. Further validation in other healthcare settings or patient groups would add greater credibility to these findings.

## Supporting information

**S1 File.**
(DOCX)

## Author Contributions

**Conceptualization:** Anchen Laubscher, Reitze N. Rodseth, Francois Retief, Adrian Saville.

**Data curation:** Anchen Laubscher, Reitze N. Rodseth, Adrian Saville.

**Investigation:** Anchen Laubscher, Reitze N. Rodseth, Adrian Saville.

**Methodology:** Anchen Laubscher, Reitze N. Rodseth, Francois Retief.

**Project administration:** Anchen Laubscher, Francois Retief.

**Software:** Reitze N. Rodseth.

**Supervision:** Adrian Saville.

**Validation:** Anchen Laubscher.

**Writing – original draft:** Anchen Laubscher.

**Writing – review & editing:** Reitze N. Rodseth, Francois Retief, Adrian Saville.

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
