## [Decision Letter · Decision Letter 0]

24 Sep 2024

PONE-D-23-29838How value perspectives influence decision-making in the South African private healthcare sector: a cross-sectional comparative studyPLOS ONE

Dear Dr. Laubscher,

Thank you for submitting your manuscript to PLOS ONE. After careful consideration, we feel that it has merit but does not fully meet PLOS ONE’s publication criteria as it currently stands. Therefore, we invite you to submit a revised version of the manuscript that addresses the points raised during the review process.

Please address all the comments submitted by reviewers.comment from the editor: please give the editor the clarification on how your paper " The Influence of Value Perspectives on Decision-Making in the South African Private Healthcare Sector" differ from the current paper" **How value perspectives influence decision-making in the South African private healthcare sector: a cross-sectional comparative study"**

Please submit your revised manuscript by Nov 08 2024 11:59PM.  If you will need more time than this to complete your revisions, please reply to this message or contact the journal office at plosone@plos.org. Please include the following items when submitting your revised manuscript:A rebuttal letter that responds to each point raised by the academic editor and reviewer(s). You should upload this letter as a separate file labeled 'Response to Reviewers'.A marked-up copy of your manuscript that highlights changes made to the original version. You should upload this as a separate file labeled 'Revised Manuscript with Track Changes'.An unmarked version of your revised paper without tracked changes. You should upload this as a separate file labeled 'Manuscript'.

We look forward to receiving your revised manuscript.

Kind regards,

Josue Mbonigaba, Ph.D

Academic Editor

PLOS ONE

Journal requirements: 1. When submitting your revision, we need you to address these additional requirements. Please ensure that your manuscript meets PLOS ONE's style requirements, including those for file naming. The PLOS ONE style templates can be found at https://journals.plos.org/plosone/s/file?id=wjVg/PLOSOne_formatting_sample_main_body.pdf and https://journals.plos.org/plosone/s/file?id=ba62/PLOSOne_formatting_sample_title_authors_affiliations.pdf. 2. Note from Emily Chenette, Editor in Chief of PLOS ONE, and Iain Hrynaszkiewicz, Director of Open Research Solutions at PLOS: Did you know that depositing data in a repository is associated with up to a 25% citation advantage (https://doi.org/10.1371/journal.pone.0230416)? If you’ve not already done so, consider depositing your raw data in a repository to ensure your work is read, appreciated and cited by the largest possible audience. You’ll also earn an Accessible Data icon on your published paper if you deposit your data in any participating repository (https://plos.org/open-science/open-data/#accessible-data). 3. We note that you have indicated that there are restrictions to data sharing for this study. For studies involving human research participant data or other sensitive data, we encourage authors to share de-identified or anonymized data. However, when data cannot be publicly shared for ethical reasons, we allow authors to make their data sets available upon request. For information on unacceptable data access restrictions, please see http://journals.plos.org/plosone/s/data-availability#loc-unacceptable-data-access-restrictions.  Before we proceed with your manuscript, please address the following prompts: a) If there are ethical or legal restrictions on sharing a de-identified data set, please explain them in detail (e.g., data contain potentially identifying or sensitive patient information, data are owned by a third-party organization, etc.) and who has imposed them (e.g., a Research Ethics Committee or Institutional Review Board, etc.). Please also provide contact information for a data access committee, ethics committee, or other institutional body to which data requests may be sent. b) If there are no restrictions, please upload the minimal anonymized data set necessary to replicate your study findings to a stable, public repository and provide us with the relevant URLs, DOIs, or accession numbers. Please see http://www.bmj.com/content/340/bmj.c181.long for guidelines on how to de-identify and prepare clinical data for publication. For a list of recommended repositories, please see https://journals.plos.org/plosone/s/recommended-repositories. You also have the option of uploading the data as Supporting Information files, but we would recommend depositing data directly to a data repository if possible. Please update your Data Availability statement in the submission form accordingly. 4. In the online submission form, you indicated that [The datasets used and/or analysed during the current study are available from the corresponding author on reasonable request.]. All PLOS journals now require all data underlying the findings described in their manuscript to be freely available to other researchers, either 1. In a public repository, 2. Within the manuscript itself, or 3. Uploaded as supplementary information.This policy applies to all data except where public deposition would breach compliance with the protocol approved by your research ethics board. If your data cannot be made publicly available for ethical or legal reasons (e.g., public availability would compromise patient privacy), please explain your reasons on resubmission and your exemption request will be escalated for approval. 5. Your ethics statement should only appear in the Methods section of your manuscript. If your ethics statement is written in any section besides the Methods, please move it to the Methods section and delete it from any other section. Please ensure that your ethics statement is included in your manuscript, as the ethics statement entered into the online submission form will not be published alongside your manuscript.  6. Please include captions for your Supporting Information files at the end of your manuscript, and update any in-text citations to match accordingly. Please see our Supporting Information guidelines for more information: http://journals.plos.org/plosone/s/supporting-information. 

Additional Editor Comments:

Please address all the comments submitted by reviewers.

comment from the editor: please give the editor the clarification on how your paper " The Influence of Value Perspectives on Decision-Making in the South African Private Healthcare Sector" differ from the current paper" **How value perspectives influence decision-making in the South African private healthcare sector: a cross-sectional comparative study"**

Reviewers' comments:

Reviewer's Responses to Questions

**Comments to the Author**

1. Is the manuscript technically sound, and do the data support the conclusions?

Reviewer #1: Partly

Reviewer #2: Partly

2. Has the statistical analysis been performed appropriately and rigorously? 

Reviewer #1: No

Reviewer #2: No

3. Have the authors made all data underlying the findings in their manuscript fully available?

Reviewer #1: No

Reviewer #2: Yes

4. Is the manuscript presented in an intelligible fashion and written in standard English?

Reviewer #1: Yes

Reviewer #2: Yes

5. Review Comments to the Author

Reviewer #1: MAJOR COMMENTS

Abstract

1. The abstract should have an introductory sentence, followed by the objectives and gaps identified. This will make the content of the background under the Abstract section meaningful. Also, include the statistical figures for the results sub-section, especially the significant findings. The formula for the relationship stated under the ‘Result’ sub-section can be written as:

VCI = (Outcome ÷ Cost) χ Patient experience

instead of expressing it as the

Value Care Index (VCI): = 

cost

Kindly rewrite the entire abstract and remove unambiguous information from the abstract. Let it be concise and meaningful.

Introduction

2. The introduction is very short and did not include much vital information on the subject matter. The introduction is the place to review other conclusions on the topic. The introduction section should introduce past findings to those who might not have that expertise. The introduction should have steps such as: introduce the topic. The first phase of the introduction is to tell the reader what the topic is and why it's interesting or important by describing the background; establishing the research problem; specifying the objective(s); and mapping out the paper. Also, remember that there are general phases that are associated with writing an introduction: stating the intent of the study, outlining the key characteristics of the study, describing important results, and giving a brief overview of the structure of the paper. Note that, the introduction of this study should follow the ‘Funnel Shape’ narrative method to scope and review existing studies across the worldwide and the study setting. This will illustrate the trends, patterns and prevalence rates of the outcome of interests of this study. Kindly revise.

3. Where are the gaps identified in this study? What has the previous studies have mentioned as regards the outcome(s) of this study? What is the current study saying on the current situation of your study outcome(s)? Always remember that a gap in a research study is an unanswered question or unresolved problem in a field, which reflects a lack of existing research in that space or it can exist when there is a concept or new idea that hasn't been studied at all, when the existing research is outdated or insufficient, or when there is a disagreement, contextual or methodological issue among the researchers. A gap in a research study indicates an opportunity for further investigation and contribution to the scientific knowledge. Kindly revise.

4. Where are the main objective of this study? Where are the specific objectives of this study? Kindly state them clearly.

5. The Keywords are so funny in nature…you have to follow the Journal format of presenting Keywords for your study. However, I may suggest the Keywords should be: Clinical outcome, patient experience, healthcare, South Africa, value perspectives, instead of you have below:

Keywords - Value of care; clinical outcome; cost of clinical event; patient experience; value perspectives; private healthcare; South Africa

6. Kindly re-write the formula for the relationship stated throughout the entire manuscript as:

VCI = (Outcome ÷ Cost) χ Patient experience

instead of expressing it as the

Value Care Index (VCI): = 

cost

Kindly revise the formula throughout the entire manuscript.

Materials and methods

7. Give a detailed description of the Study setting and design - First, give us a detailed geographical description of the study setting you carried out this study. What is the justification for the choice of the study setting? Also, second, what is the type of design you employed in this study? Kindly address this.

8. Sampling methods – kindly describe the type of sampling procedures you initiated when collecting the data via online survey platforms. There should be explanation on the study method and, detailed explanations on the methods on the specific tools and procedures use in collecting and analyzing the data should be clarified. Give a justification for the choice of sampling method employed. Kindly address this.

9. Study population and Sampling technique – describe your study population? How did you arrived at reaching out to the population target? Give justification of the choice of these procedures you employed? How did you select them? Give us an adequate and precise procedures you used in getting your population. Discuss in detail both of them with a separate section paragraph for each of them. Insert justification for both differently. Kindly address this.

10. Variable measurements (Outcome variable or Dependent variable) – Where is the outcome variable? A dependent variable is the variable that changes as a result of the independent variable (independent or explanatory variables or factors) manipulation. It's the outcome you're interested in measuring, and it “depends” on your independent variable. In statistically analysis, dependent variables are also called ‘response variables’ (they respond to a change in another variable). Therefore, specify your outcome variable very well and show us how it is going to be measured. Is it a binary outcome variable or what? This section should discuss the measurements of the outcome variable or dependent variable(s). Importantly, Kindly address this.

11. Explanatory variable (factors) or independent variable (factors): Where are the explanatory variable listed? You did not indicate your independent variables you used in this study. I know that you cannot have the dependent variable without the independent variable (factors) when running the statistical analysis. Insert the independent variables and how they are going to be measured (References may be cited). Define all your Explanatory variables (factors) or Independent variables (factors) and how they were measured in your study. This is very important, as it will affect the process and approach of statistical analysis. Kindly address this.

Statistical analysis –

12. This statement in blue ink [Descriptive statistics were used to characterise the study sample and tests of mean differences (e.g., independent samples t tests and analyses of variance [ANOVA]) were used to assess whether value perspective differences existed between consumer and provider groups, as well as to assess value perspective differences as the severity of surgical and medical scenarios increased – Page 9] – is not accurate. Descriptive statistics are not tests of mean differences; rather descriptive statistics allow you to characterize your data based on its properties, and there are four major types of descriptive statistics, which include the following: 1. Measures of Frequency (Count, Percent, and Frequency); 2. Measures of Central Tendency (Mean, Median, and Mode); 3. Measures of Dispersion or Variation (Range, Variance, and Standard Deviation); and 4. Measures of Position (Percentile Ranks, and Quartile Ranks)

13. State and describe the type of statistical analysis suitable for each objective and how they will be analysed in this study.

14. Did you employ the use of parametric statistical tests or non-parametric statistical tests?

15. Did you use the statistical methods in the data analysis? The statistical methods include descriptive statistics (which summarizes data using indexes such as mean and median) and inferential statistics (which draw conclusions from data using statistical tests such as student's t-test).

16. Also, there should be step-by-step description of the analysis by objectives shown in the statistical analysis’ sub-section. For instance, objective one on respondents’ demographic characteristics was analyzed using univariate analysis (Percentage and frequency including graphs and tables). Objective two measure the associations between the outcome variable and explanatory variables by using the bivariate analysis (Chi-Square or/and Pearson correlation coefficient or non-parametric statistical methods). Objective three measures the relationship influence between the variables and the predictors of the outcome variable on the explanatory factors, when other factors are kept constant (for instance, when using Logistics regression, if and only if the outcome variable is binary, Yes = 1 and No = 0 or the use of non-parametric statistical methods), otherwise, multiple régression analysis, etc.

This section above must be revised as it is very important. Also, include the models and equations of the bivariate and inferential statistical analysis in your discussion under the section of the statistical analyses. Kindly address this question.

17. This statement in blue is not accurate [Parametric assumptions were evaluated using Kolmogarov-Smirnov and Shapiro-Wilk tests]. Kolmogarov-Smirnov and Shapiro-Wilk tests are not parametric or have parametric assumptions, rather they are non-parametric tests with its assumptions.

18. This statement in blue is not accurate [To examine the association between the variables of interest, univariate analyses using Pearson product-moment correlations were conducted]. It is only Inferential statistics that are involved in examining the associations (use of Chi-square tests) or relationships (Pearson product-moment correlations or Spearman rank).

19. I suggest you put more efforts in understanding the principles of statistical analytical methods. Therefore, kindly explain the each of the statistical methods you want to use and give reasons why you want to use it. For instance, kindly explain how and why you are using the following as you have previously mentioned in the manuscript:

1. Non-parametric statistical test - Kolmogarov-Smirnov test: Assumptions, How and why?

2. Non-parametric statistical test - Shapiro-Wilk test: Assumptions, How and why?

3. Inferential statistical test - Independent sample t-test: Assumptions, How and why?

4. Inferential statistical test - Cohen’s d effect size: Assumptions, How and why?

5. Inferential statistical test - One way between-groups ANOVAs: Assumptions, How and why?

20. State and explain the type of analysis you will used to explain each objective of this study

Description of the statistical methods

Kolmogarov-Smirnov test

The Kolmogorov-Smirnov test is used to test the null hypothesis that a set of data comes from a Normal distribution. The Kolmogorov Smirnov test produces test statistics that are used (along with a degrees of freedom parameter) to test for normality. Here we see that the Kolmogorov Smirnov statistic takes value . One-sample K-S test or goodness of fit test was developed by Andrey Nikolayevich Kolmogorov in 1933. Its purpose is to compare the overall shapes of two sample distributions. Two-sample K-S test was developed by Nikolai Smirnov in 1939. Its purpose is to compare one sample to a known statistical distribution. The interpretation of the result, where the p-value, tells you how likely these samples comes from the exact same distribution. When obtained, the p-value can be compared with a threshold call statistical significance (e.g. . 05), if the P-Value is smaller, we can reject the null hypotheses.

Shapiro-Wilk test

The Shapiro–Wilk test can be used to decide whether or not a sample fits a normal distribution, and it is commonly used for small samples. The interpretation of the findings is if the value of the Shapiro-Wilk Test is greater than 0.05, the data is normal, however, if it is below 0.05, the data significantly deviate from a normal distribution. The null hypothesis for a Shapiro Wilk test is that there is no difference between your distribution and a normal distribution. The alternative hypothesis is that there is a difference. If your p value is less than 0.05, which it is, then you reject the null hypothesis and conclude that your data is non-normal.

Independent sample t-test

The independent t-test, also called the two sample t-test, independent-samples t-test or student's t-test, is an inferential statistical test that determines whether there is a statistically significant difference between the means in two unrelated groups. The Independent Samples t Test compares the means of two independent groups in order to determine whether there is statistical evidence that the associated population means are significantly different. The Independent Samples t Test is a parametric test. Paired-samples t tests compare scores on two different variables but for the same group of cases; independent-samples t tests compare scores on the same variable but for two different groups of cases. What are the two types of independent t tests? The independent samples t-test comes in two different forms: 1. the standard Student's t-test, which assumes that the variance of the two groups are equal; and 2. Welch's t-test, which is less restrictive compared to the original Student's test. The interpretation of the Independent Samples t Test is If the p-value is less than your significance level (e.g., 0.05), you can reject the null hypothesis. The difference between the two means is statistically significant. Your sample provides strong enough evidence to conclude that the two population means are not equal.

Cohen’s d effect size

There are dozens of measures of effect sizes. The most common effect sizes are Cohen's d and Pearson's r. Cohen's d measures the size of the difference between two groups while Pearson's r measures the strength of the relationship between two variables. Interpreting Cohen’s d A commonly used interpretation is to refer to effect sizes as small (d = 0.2), medium (d = 0.5), and large (d = 0.8) based on benchmarks suggested by Cohen (1988). However, these values are arbitrary and should not be interpreted rigidly (Thompson, 2007). What if Cohen's d is greater than 1? A Cohen's d of 1.000 indicates that the means of the two groups differ by 1.000 pooled standard deviation (or one z-score). A Cohen's d of 2.00 indicates that the means of two groups differ by 2.000 pooled standard deviations, and so on. How do you interpret Cohen's d example? A large Cohen's d indicates the mean difference (effect size = signal) is large compared to the variability (noise). For example, if Group A's Mean = 12 and Group B's Mean = 8, and the pooled standard deviation is 2, Cohen's d equals the following: The mean difference is twice the variability.

One way between-groups ANOVAs

ANOVA is categorised into two: 1. between within groups ANOVA; and 2. between groups ANOVA. The within group variation is also called random, unexplained, residual, or variation due to error, and the Residual or within-group variation is assumed to follow the normal, Gaussian, distribution. Also, the between group variation is due to the factor, or explained by the factor. A One-Way Between-Subjects ANOVA compares the means between more than two independent groups, such as comparing the difference between groups A, B and C. If your data only has two groups such as Male/Female or Present/Absent you should consider the Independent-Samples t-Test. In ANOVA, F is the ratio of the error mean square to the model mean square. The P-value is the probability, if the means of the various populations are equal, of a value of F greater than what you observed. By the way, your question is exactly why I dislike terms like P-value*. It doesn't say what it means.

21. Where there missing questionnaires after collation of the questionnaire from the field? If any, how did your account for missing variables or non-responses from the respondents? Kindly look into this.

22. Where is the Ethical approval number assigned to this study research? Please, kindly include the Ethical Approval Reference number.

23. Where is the “Declaration of Helsinki Ethical principles” of this study? What ethical principles did you employed? Did you follow the set of ethical principles. Where is the informed consent form in research? How did you address these factors associated with research such as in ensuring in the protection of participants’ rights, ensuring that participants understand the nature and purpose of the research, the risks and benefits of participating, and their rights as participants whether to participate or decline at any time they feel not to. Kindly address this.

24. Was informed consent forms given to the respondents during the data collection via online media outlets you used?

Results

25. There is no table for the demographics for the women respondents. Kindly insert the socio-demographics features of your study respondents.

26. Kindly report the results according to the objectives by using the objective as a theme heading. Show the type of analysis employed and elaborate on the interpretation of the study findings. All the interpretations for the Tables should be included in the results and Table legend should be carefully and appropriately inserted in the main research. Interprete and discuss the relevant and significant variables and make it precise so that it doesn’t lose its meaning. Your readers will see the tables with the results. Kindly address these concerns mentioned. Therefore, kindly revise. For instance, see below: - Table 1 – Demographic Characteristics of Respondents; - Table 2 – Descriptive statistical analysis; - Table 3 – Inferential statistical analysis (independent t-sample test); - Table 4 – Non-parametric analysis; and - Other relevant tables (if any). Kindly address this.

27. Where is Figure 1?

28. All the findings for this study must be reported according to the objectives of this study.

Discussion

29. The discussion section is one of the final parts of a research paper, in which an author describes, analyzes, and interprets their findings. They explain the significance of those results and tie everything back to the research question(s). The discussion section is where you delve into the meaning, importance, and relevance of your results. It should focus on explaining and evaluating what you found, showing how it relates to your literature review and paper or dissertation topic, and making an argument in support of your overall conclusion. Therefore, let your discussion aspects focused on the interpretation of your findings and relate to literature whether it corroborates or not, with existing studies, alongside with in-citation of recent references. Let each paragraph relates the objective findings. Discuss in assertion or not with study’s findings from other studies. Kindly advise.

Conclusion

30. The conclusion should stem from summing up your research paper (by the following steps - restate your research topic, restate the thesis, summarize the main points, state the significance or results, and conclude your thoughts). Always have it in mind that a conclusion is not merely a summary of the main topic(s) covered or a re-statement of your research problem, but a synthesis of key points and, if applicable, where you should recommend new areas for future research.

Strengths of this Study -

31. Where are the Strengths of this Study? Please state them, after the discussion with the sub-Section heading: Strengths and Limitations of this Study after the Discussion Section. Add the limitations of the study after the strengths of the study is included.

Implications of your findings –

32. What are the implications of your findings across other countries with lower prevalence of your outcome of interest? What is the contribution of this study to the existing one, especially in sub-Saharan African or developing countries?

Recommendations

33. PLEASE SEE THE PDF SENT TO THE AUTHOR, ITS COMPLETED.

Reviewer #2: The concept of integrating patient experience, clinical outcomes, and cost to measure healthcare value, as proposed in this study through the Value Care Index (VCI), aligns closely with the well-established Triple Aim framework by the Institute for Healthcare Improvement (IHI). Introduced by Berwick et al. in 2008 (Berwick DM, Nolan TW, Whittington J. The Triple Aim: Care, Health, and Cost. Health Aff (Millwood). 2008;27(3):759-69), the IHI Triple Aim already emphasizes the balance between population health outcomes, per capita costs, and patient experience.

Given that similar work has been done, I suggest the authors clarify how their approach offers novel insights beyond the Triple Aim. It would also be helpful for the authors to address the existing body of research and explain how their VCI differs from or improves upon this widely recognized framework.

Feedback on Introduction Structure:

The paragraph that outlines the study's objectives is important, but it could be better placed after a more detailed discussion of the existing literature and research gap. I recommend:

Expanding the background on patient experience, clinical outcomes, and event cost with appropriate citations.

Clarifying the research gap by explaining why the relationship between patient experience and value is less understood.

Moving the research objectives to a separate paragraph at the end of the introduction, which will provide a clearer transition from the background to the aims of the study.

While the manuscript is well-structured and covers a critical topic, I would like to bring your attention to a minor grammatical issue. On page 6, the sentence reads: “Data were collect from 1 to 30 September 2017.” The correct phrase should be: “Data were collected from 1 to 30 September 2017.”

While the "Materials and Methods" section is thorough and detailed, which is commendable, the "Introduction" and "Literature Survey" sections are comparatively underdeveloped. To set the context of the study more effectively, the introduction should include a more comprehensive literature review that evaluates similar works done in the area. This would help in framing the research gap more explicitly and support the study's objectives.

I suggest the following improvements:

Expand the literature review: Include more studies that explore the relationship between patient experience, clinical outcomes, and event costs in healthcare, particularly in similar contexts. This would enhance the academic rigor and provide a stronger foundation for the study.

Clarify the research gap: Explicitly state what existing studies have missed and why this study is essential in the South African healthcare context.

Condense some parts of the Materials and Methods: While the detailed methodology is valuable, some minor adjustments to condense certain descriptions may help balance the overall structure of the paper.

Review Comment:

While it is understood that the questionnaire has already been administered, the choice of clinical scenarios in Table 1 raises some concerns regarding how non-healthcare professionals (non-HCPs) would relate to and comprehend the descriptions, particularly for complex procedures they may not have experienced (e.g., "liver cancer that has spread" or "severe shock as a complication of a heart attack"). This could have affected the validity of their responses.

To address this, I suggest the authors:

Provide a justification for the selection of these specific clinical examples. Why were these particular scenarios chosen, and on what basis were they considered relatable to non-HCPs?

Clarify the rationale for expecting non-HCPs to rate their expectations for procedures they may not have personal experience with.

Acknowledge the potential limitation this introduces and discuss how it may impact the interpretation of the results. For instance, did the authors anticipate this issue and take any measures to mitigate it?

The "Data Analysis" section provides a good overview of the statistical techniques employed. However, I recommend a few enhancements for clarity and comprehensiveness:

Clarify alignment with research questions: While you mention the statistical tests used, it would strengthen the section to explicitly link these tests to the specific research questions or hypotheses they were designed to address (e.g., which tests were used to assess differences in value perspectives across severity levels).

Missing data handling: Please clarify if any missing data were present in the dataset and how this was handled in the analysis (e.g., listwise deletion, imputation).

Effect size interpretation: You mention Cohen’s d for effect size; it might be useful to briefly state how the effect sizes were interpreted (small, medium, large), as this will help readers understand the practical significance of the findings.

Assumption checks: The section on parametric assumptions (Kolmogorov-Smirnov and Shapiro-Wilk tests) is helpful. However, consider noting how any violations of these assumptions were handled (e.g., transformations or non-parametric alternatives).

Review Comment:

The proposed Value Care Index (VCI) offers a compelling way to incorporate patient experience into the healthcare value equation, and the consistent valuation of patient experience across scenarios supports this idea. However, as this is a significant claim, it might benefit from additional refinement and validation using a broader set of data or more diverse clinical contexts. I suggest the following:

Expand on the rationale for the equation: It would strengthen the proposal to further explain why this specific formulation of the VCI was chosen and how it compares to other models of healthcare value measurement.

Consider more data for validation: While the current data provides a strong starting point, further validation across different healthcare settings or additional patient groups could lend more credibility to the equation. This could be addressed as a future research direction.

Discuss limitations: Acknowledging the potential limitations of applying the VCI broadly and proposing future studies to refine the coefficients and weightings would show a balanced view without undermining the innovation of this idea.

Review Comment:

In addition to the limitations already mentioned, I recommend that the authors address the geographical limitations of the study. The findings, particularly those related to value perspectives in healthcare, may be specific to the South African private healthcare context and may not be directly applicable to other healthcare systems globally. Given that healthcare values can vary by country and region, acknowledging this limitation will provide a clearer understanding of how widely the results can be generalized. I also suggest discussing the potential differences in value perspectives between healthcare systems globally and how these might impact the applicability of the proposed Value Care Index (VCI).

Review Comment:

The section on aligning healthcare around the goal of providing patients with the highest possible value of care is crucial, especially in the South African private healthcare context. However, I believe it can be improved by more clearly highlighting the study's specific findings and their implications. For example, the fact that patient experience consistently holds a higher perceived value than event cost across various scenarios should be emphasized. Additionally, the differences in value perception between consumers and providers regarding clinical outcomes, patient experience, and costs should be made more explicit, as these are key insights that can inform value-driven healthcare strategies.

I recommend revising the paragraph to focus more on the findings and their relevance to healthcare leaders in South Africa. This will make the study’s contributions clearer and more impactful.

6. PLOS authors have the option to publish the peer review history of their article (what does this mean?). If published, this will include your full peer review and any attached files.

Reviewer #1: **Yes: **Dr. Monica Ewomazino Akokuwebe

Reviewer #2: **Yes: **Thamburaj Anthuvan

---

## [Author Response · Author response to Decision Letter 0]

13 Nov 2024

How value perspectives influence decision-making in the South African private

healthcare sector: a cross-sectional comparative study

Response to reviewers. 

Reviwer one

 Abstract 

1. The abstract should have an introductory sentence, followed by the objectives and gaps identified. This will make the content of the background under the Abstract section meaningful. Also, include the statistical figures for the results sub-section, especially the significant findings. The formula for the relationship stated under the ‘Result’ sub-section can be written as: VCI = (Outcome ÷ Cost) χ Patient experience instead of expressing it as the Value Care Index (VCI): 𝑉𝐶𝐼 = 𝑜𝑢𝑡𝑐𝑜𝑚𝑒 𝑥 𝑝𝑎𝑡𝑖𝑒𝑛𝑡 𝑒𝑥𝑝𝑒𝑟𝑖𝑒𝑛𝑐𝑒./cost. Kindly rewrite the entire abstract and remove unambiguous information from the abstract. Let it be concise and meaningful.

Response

We have rewritten the abstract to incorporate the recommendations made. We have unfortunately not been able to include the figures for the significant statistical findings into the abstract results. Because of the multiple comparisons – three factors (cost, clinical outcome, patient experience) by two categories (patients and providers) – incorporating there pushes the word count of the abstract well beyond the allowed 300 words. 

2. The introduction is very short and did not include much vital information on the subject matter. The introduction is the place to review other conclusions on the topic. The introduction section should introduce past findings to those who might not have that expertise. The introduction should have steps such as: introduce the topic. The first phase of the introduction is to tell the reader what the topic is and why it's interesting or important by describing the background; establishing the research problem; specifying the objective(s); and mapping out the paper. Also, remember that there are general phases that are associated with writing an introduction: stating the intent of the study, outlining the key characteristics of the study, describing important results, and giving a brief overview of the structure of the paper. Note that, the introduction of this study should follow the ‘Funnel Shape’ narrative method to scope and review existing studies across the worldwide and the study setting. This will illustrate the trends, pattens and prevalence rates of the outcome of interests of this study. Kindly revise. 

3. Where are the gaps identified in this study? What has the previous studies have mentioned as regards the outcome(s) of this study? What is the current study saying on the current situation of your study outcome(s)? Always remember that a gap in a research study is an unanswered question or unresolved problem in a field, which reflects a lack of existing research in that space or it can exist when there is a concept or new idea that hasn't been studied at all, when the existing research is outdated or insufficient, or when there is a disagreement, contextual or methodological issue among the researchers. A gap in a research study indicates an opportunity for further investigation and contribution to the scientific knowledge. Kindly revise. 

4. Where are the main objective of this study? Where are the specific objectives of this study? Kindly state them clearly. 

Reponses to 2, 3 and 4.

Thank you, we have expanded the introduction to consider these recommendations. 

5. The Keywords are so funny in nature…you have to follow the Journal format of presenting Keywords for your study. However, I may suggest the Keywords should be: Clinical outcome, patient experience, healthcare, South Africa, value perspectives, 

Response

Thank you we have made use of these key words

6. Kindly re-write the formula for the relationship stated throughout the entire manuscript as: VCI = (Outcome ÷ Cost) χ Patient experience Kindly revise the formula throughout the entire manuscript.

Response

 Thank you – we have made this change.

7. Give a detailed description of the Study setting and design - First, give us a detailed geographical description of the study setting you carried out this study. What is the justification for the choice of the study setting? Also, second, what is the type of design you employed in this study? Kindly address this. 

Response: 

 This information was provided in the materials and methods section. The text reads as follows:

“This study was a non-experimental, cross-sectional, comparative study between healthcare consumer and provider groups within the private healthcare sector in South Africa. The industry in which the study was conducted was the South African healthcare sector, specifically, the private sector where payment is made for medical services rendered. Clinicians are not employed by private healthcare institutions and operate as independent practitioners in these facilities, as dictated by the Health Professionals Council of South Africa (HPCSA). Patients are either insured by subscribing to and purchasing of medical aid cover or they pay for private medical care by means of cash at point of care.”

8. Sampling methods – kindly describe the type of sampling procedures you initiated when collecting the data via online survey platforms. There should be explanation on the study method and, detailed explanations on the methods on the specific tools and procedures use in collecting and analyzing the data should be clarified. Give a justification for the choice of sampling method employed. Kindly address this. 

Response:

 This information was provided in the second part of the materials and methods section. The text reads as follows: 

“A non-probability sampling technique was used to collect data and included convenience,

purposive and snowball sampling strategies [6]. The first author (AL) utilised her professional and personal network to collect data from the two sample groups using various social media platforms. The purposive strategy was employed to ensure a high number of clinician participants across a wide spectrum of subspecialties and registration categories with the HPCSA. Included in the sample were doctors working in both administrative and clinical capacities. Snowball sampling was used to leverage the networks available to the researcher. Participants were asked to distribute the survey to other potentially willing individuals both when they received an invitation and following completion of the survey.”

9. Study population and Sampling technique – describe your study population? How did you arrived at reaching out to the population target? Give justification of the choice of these procedures you employed? How did you select them? Give us an adequate and precise procedures you used in getting your population. Discuss in detail both of them with a separate section paragraph for each of them. Insert justification for both differently. Kindly address this. 

Response: 

 This information was provided in the materials and methods. It reads as follows: 

This study was a non-experimental, cross-sectional, comparative study between healthcare consumer and provider groups within the private healthcare sector in South Africa. The industry in which the study was conducted was the South African healthcare sector, … 

A non-probability sampling technique was used to collect data and included convenience, purposive and snowball sampling strategies [6]. The first author (AL) utilised her professional and personal network to collect data from the two sample groups using various social media platforms. The purposive strategy was employed to ensure a high number of clinician participants across a wide spectrum of subspecialties and registration categories with the HPCSA. Included in the sample were doctors working in both administrative and clinical capacities….

… Data were collected through a self-administered online survey, the Value Perspectives Survey, using the survey platform Qualtrics (full survey available in Appendix 1) and were collect from 1 to 30 September 2017. The survey link directed the participant to a cover letter after which consent was obtained. Participants were then directed to either the patient or doctor section of the survey. To ensure a between-subjects design, a participant could not compete the survey from both consumer and provider perspectives. Should a participant work in the healthcare industry as a healthcare worker in any category other than that of a doctor, they were required to complete the survey from a patient perspective. 

10. Variable measurements (Outcome variable or Dependent variable) – Where is the outcome variable? A dependent variable is the variable that changes as a result of the independent variable (independent or explanatory variables or factors) manipulation. It's the outcome you're interested in measuring, and it “depends” on your independent variable. In statistically analysis, dependent variables are also called ‘response variables’ (they respond to a change in another variable). Therefore, specify your outcome variable very well and show us how it is going to be measured. Is it a binary outcome variable or what? This section should discuss the measurements of the outcome variable or dependent variable(s). Importantly, Kindly address this. 

Response

 The outcome was specifically provided in the materials and methods section and formed the basis of the analysis we conducted. It read as follows: 

“After obtaining basic demographic information the survey presented clinical scenarios aimed at examining the participants’ value perspectives. They were asked to distribute 100 points between clinical outcomes, event cost, and patient experience according to their perceived relative importance.”

To make this clearer to the reader we have added clarifying information. The section now reads as follows: 

“After obtaining basic demographic information the survey presented clinical scenarios aimed at examining the participants’ value perspectives – the primary focus of this study. They were asked to distribute 100 points between clinical outcomes, event cost, and patient experience according to their perceived relative importance. For example, a participant evaluating a clinical scenario where they were undergoing surgery for a hernia repair could decide to allocate 30 of the available 100 points to clinical outcome, 50 points to event cost, and 20 to patient experience.”

11. Explanatory variable (factors) or independent variable (factors): Where are the explanatory variable listed? You did not indicate your independent variables you used in this study. I know that you cannot have the dependent variable without the independent variable (factors) when running the statistical analysis. Insert the independent variables and how they are going to be measured. Define all your Explanatory variables (factors) or Independent variables (factors) and how they were measured in your study. This is very important, as it will affect the process and approach of statistical analysis. Kindly address this. 

Response:

 This information was provided in multiple places in both the introduction, and formally in the materials and methods section. The two independent variables, or groups of interest, were healthcare consumers and healthcare providers. This was stated in the following sections:

Introduction 

- “Further to exploring the role of Patient’s Experience in Value of Care, the purpose of the study is to obtain a deeper understanding of the differences in value perspectives between healthcare consumer and provider groups”

- “This study, a quantitative review of primary data, aims to gain insight into the value perspectives of consumers and providers in the South African private healthcare context and to examine if these value perspectives differ between the two groups. Both providers and consumers from the South African private sector were included in this study.”

- “The study explored the differences between the relative importance of the three factors (Clinical Outcome, Cost of Clinical Event, and Patient’s Experience) that make up the Value of Care for both consumers and providers, in general and specifically in terms of medical and surgical procedures.”

- “The objective of this study was to … understand how relative importance between the three factors might differ between consumer and provider groups as the clinical severity of the scenarios changed.”

Material and methods

- “This study was a non-experimental, cross-sectional, comparative study between healthcare consumer and provider groups within the private healthcare sector in South Africa.”

Data analysis

- “Descriptive statistics were used to characterise the study sample and tests of mean differences (e.g., independent samples t tests and analyses of variance [ANOVA]) were used to assess whether value perspective differences existed between consumer and provider groups”

The definition of the consumer and provider groups were provided in the materials and methods section. 

12. This statement in blue ink [Descriptive statistics were used to characterise the study sample and tests of mean differences (e.g., independent samples t tests and analyses of variance [ANOVA]) were used to assess whether value perspective differences existed between consumer and provider groups, as well as to assess value perspective differences as the severity of surgical and medical scenarios increased – Page 9] – is not accurate. Descriptive statistics are not tests of mean differences; rather descriptive statistics allow you to characterize your data based on its properties, and there are four major types of descriptive statistics, which include the following: 1. Measures of Frequency (Count, Percent, and Frequency); 2. Measures of Central Tendency (Mean, Median, and Mode); 3. Measures of Dispersion or Variation (Range, Variance, and Standard Deviation); and 4. Measures of Position (Percentile Ranks, and Quartile Ranks) 

Response

 Thank you for this comment. 

Our sentence read as follows: “Descriptive statistics were used to characterise the study sample and tests of mean differences (e.g., independent samples t tests and analyses of variance [ANOVA]) were used to assess whether value perspective differences existed between consumer and provider groups, as well as to assess value perspective differences as the severity of surgical and medical scenarios increased.”

As the sentence says, descriptive statistics were used to characterize the study sample. 

“Descriptive statistics were used to characterise the study sample.”

And then tests of mean difference were used to assess the differences between the two groups. 

“…and tests of mean differences (e.g., independent samples t tests and analyses of variance [ANOVA]) were used to assess whether value perspective differences existed.”

This formulation is standard when describing basic statistical methods. 

13. State and describe the type of statistical analysis suitable for each objective and how they will be analysed in this study. 

Response: 

 We refer the reviewer to the section of the manuscript titled “Data analysis”, and specifically the section that reads:

“ … tests of mean differences (e.g., independent samples t tests and analyses of variance [ANOVA]) were used to assess whether value perspective differences existed between consumer and provider groups, as well as to assess value perspective differences as the severity of surgical and medical scenarios increased.”

“… To examine the association between the variables of interest, univariate analyses using Pearson product-moment correlations were conducted. Independent samples t-tests were conducted to assess whether provider and consumer groups weighted value perspectives differently across the general value baseline and the surgical, and medical scenarios. Cohen’s d effect sizes were calculated to determine practical differences between the weighted perspectives. One way between-groups ANOVAs were conducted to determine whether differences existed in value perspectives across severity of surgical and medical scenarios for both consumer and provider groups.”

These specifically note the outcome being analyzed, as well as the test being used. 

14. Did you employ

---

## [Editor Report · Decision Letter 1]

29 Nov 2024

PONE-D-23-29838R1How value perspectives influence decision-making in the South African private healthcare sector: a cross-sectional comparative studyPLOS ONE

Dear Dr. Laubscher,

Thank you for submitting your manuscript to PLOS ONE. After careful consideration, we feel that it has merit but does not fully meet PLOS ONE’s publication criteria as it currently stands. Therefore, we invite you to submit a revised version of the manuscript that addresses the points raised during the review process. 1) The heading "introduction" on the abstract should be replaced with " Background" 2) The sub-heading " importance of patient experience" in the introduction section of the paper is not necessary 3) Clarify the contribution of the study to related work in this area emphasizing advancement of knowledge on most recent work 4) Page 14" while it was negatively corrected" should be replaced with " while it was negatively correlated 

Kind regards,

Josue Mbonigaba, Ph.D

Academic Editor

PLOS ONE
---

## [Author Response · Author response to Decision Letter 1]

10 Dec 2024

Response to reviewers

1) The heading "introduction" on the abstract should be replaced with " Background" 

We have made this change. 

2) The sub-heading " importance of patient experience" in the introduction section of the paper is not necessary 

We have removed this sub-heading.

3) Clarify the contribution of the study to related work in this area emphasizing advancement of knowledge on most recent work.

We have added a section to the conclusion highlighting the unique contribution to the field made by this study. The section reads as follows:

“While our findings support the centrality of clinical outcomes as the key determinant in the value equation, they highlight the importance of patient experience over event cost and make the argument for its inclusion into the tradition value equation. Our study further contributes to our knowledge of this subject by describing how: 1) relative value proportions of clinical outcome, patient experience and patient event cost when determining healthcare value; 2) clinical severity impacts value assignment; 3) value judgements differ between medical and surgical clinical scenarios, and 4) healthcare consumers and providers differ in assigning value – all within the South African healthcare context.”

4) Page 14" while it was negatively corrected" should be replaced with " while it was negatively correlated 

We have corrected this to read “while it was negatively correlated”.

---

## [Editor Report · Decision Letter 2]

13 Dec 2024

How value perspectives influence decision-making in the South African private healthcare sector: a cross-sectional comparative study

PONE-D-23-29838R2

Dear Authors,

We’re pleased to inform you that your manuscript has been judged scientifically suitable for publication and will be formally accepted for publication once it meets all outstanding technical requirements.

Kind regards,

Josue Mbonigaba, Ph.D

Academic Editor

PLOS ONE

Additional Editor Comments (optional):

NA
---

## [Editor Report · Acceptance letter]

6 Jan 2025

PONE-D-23-29838R2 

PLOS ONE

Dear Dr. Laubscher, 

I'm pleased to inform you that your manuscript has been deemed suitable for publication in PLOS ONE. Congratulations! Your manuscript is now being handed over to our production team.

Kind regards, 

on behalf of

Dr. Josue Mbonigaba 

Academic Editor

PLOS ONE